# The burden of stroke and modifiable risk factors in Ethiopia: A systemic review and meta-analysis

Teshager Weldegiorgis Abate[1]*, Balew Zeleke[2‡], Ashenafi Genanew[3☉], Bidiru Weldegiorgis Abate[4☉]

1 Department of Adult Health Nursing, School of Health Science, College of Medicine and Health Science, Bahir Dar University, Bahir Dar, Ethiopia, 2 Department of Pediatric and Child Health Nursing, School of Health Sciences, College of Medicine and Health Sciences, Bahir Dar University, Bahir Dar, Ethiopia, 3 Department of Pharmacology, School of Health Sciences, College of Medicine and Health Science, Bahir Dar University, Bahir Dar, Ethiopia, 4 Department of Anesthetics, Addis Alem Hospitla, Amhara Regional Bureau, Bahir Dar, Ethiopia

☉ These authors contributed equally to this work.
‡ These authors also contributed equally to this work.
* teshagerabate9@gmail.com

**Data Availability Statement:** All relevant data are within the paper and its Supporting Information files.

**Funding:** The authors received no specific funding for this work.

## Abstract

### Background

The burden and contribution of modifiable risk factors of stroke in Ethiopia are unclear. Knowledge about this burden and modifying risk factors is pivotal for establishing stroke prevention strategies. In recent decades, the issue of lifestyle and behavioral modification is a key to improve the quality of life. The modifiable risk factors are an importance as intervention strategies aimed at reducing these factors can subsequently reduce the risk of stroke. So far, many primary studies were conducted to estimate the burden of stroke and modifiable risk factors in Ethiopia. However, the lack of a nationwide study that determines the overall pooled estimation of burden and modifiable risk factors of stroke is a research gap.

### Methods

To conduct this systemic review and meta-analysis, we are following the PRISMA checklist. Three authors searched and extracted the data from the CINAHL (EBSCO), MEDLINE (via Ovid), PubMed, EMcare, African Journals Online (AJOL), and Google scholar. The quality of the primary study was assessed using the Newcastle-Ottawa Scale (NOS) by two independent reviewers. The primary studies with low and moderate risk of bias were included in the final analysis. The authors presented the pooled estimated burden of stroke and its modifiable risk factors. The registered protocol number in PROSPERO was CRD42020221906.

### Results

In this study, the pooled burden of hemorrhagic and ischemic stroke were 46.42% (95%CI: 41.82–51.53; $I^2$ = 91.6%) and 51.40% (95%CI: 46.97–55.82; $I^2$ = 85.5%) respectively. The overall magnitude of modifiable risk factor of hypertension, alcohol consumption and

**Competing interests:** No authors have competing interests.

**Abbreviations:** DALYs, Disability-Adjusted Life-Years; NCDs, Non-Communicable Diseases; MESH, Medical Subject Headings; SNNPR, South Nations, Nationalities and People Region; NOS, Newcastle-Ottawa Scale.

dyslipidemia among stroke patients were 49% (95%CI: 43.59, 54.41), 24.96% (95CI%:15.01, 34.90), and 20.99% (95%CI: 11.10, 30.88), respectively. The least proportion of stroke recovery was in the Oromia region (67.38 (95%CI: 41.60–93.17; I2 = 98.1%). Farther more, the proportion of stroke recovery was decreased after 2017 (70.50 (56.80–84.20).

## Conclusions

In our study, more than 90% of stroke patients had one or more modifiable risk factors. All identified modifiable stroke risk factors are major public health issues in Ethiopia. Therefore, strategy is designed for stroke prevention to decrease stroke burden through targeted modification of a single risk factor, or a cluster of multiple risk factors, used on a population, community, or individual level.

## Background

Stroke remains the second leading cause of death worldwide with an annual mortality rate of 5·5 million. Fewer women (2·6 million) than men (2·9 million) have died from stroke [1, 2]. The incidence, prevalence, and mortality rate of stroke have increased worldwide, with most of the burden being in the low and middle-income countries including Ethiopia [3, 4]. Hemorrhagic stroke is responsible for more deaths and Disability-Adjusted Life-Years (DALYs). Incidence and mortality of stroke differ between countries, geographical regions, and ethnic groups [5].

Ethiopia faces the unenviable threat of a triple burden of disease: infectious diseases, Non-Communicable Diseases (NCDs), and injuries [6]. Although Ethiopia is progressing towards national health coverage, the country faces the triple burden of diseases [7]. The magnitude of stroke-related deaths in Ethiopia is 6.23% out of total deaths, and the age-adjusted death rate of stroke in the country is 89.82 per 100 000 of the population [8]. Besides, previous reports indicated that 90% of the burden of stroke is attributable to modifiable risk factors [9]. Of this, three-quarters of the stroke burden is attributable to behavioral risk factors [10]. Metabolic factors (high blood pressure, obesity, fasting plasma glucose, cardiac disorder, and total cholesterol) accounted for 72% of stroke DALYs, and behavioral factors (smoking, poor diet, and physical inactivity) accounted for 66% [11–15]. In Ethiopia, a comprehensive nationally representative study on stroke burden and its modifiable risk factor are lacking. Thus, this study aimed to determine the overall pooled burden and its modifiable risk factors of stroke in Ethiopia.

## Methods and analysis

### Protocol design and registration

A systematic review with a meta-analysis of published and unpublished observational studies was incorporated to assess the burden of stroke and its modifiable risk factors in Ethiopia. To develop this systemic review and the meta-analysis, the authors used the Preferred Reporting Items for Systematic Review and Meta-analysis Protocol (PRISMA-P) [16, 17] and Meta-analysis of Observational Studies in Epidemiology (MOOSE) guideline statement [18]. This systemic review and meta-analysis protocol was registered in the International Registration of Systems Reviews (PROSPERO) with CRD 42020221906.

## Eligibility criteria

The eligibility of the study was determined using the following criteria: (1) all facility-based observational studies; (2) all studies conducted in Ethiopia; (3) all studies reporting either the magnitude of any subtypes of stroke or rate of improvement at discharge and modifiable risk factors; and both published and unpublished studies. On the other hand, the authors excluded the following: anonymous reports, case reports, qualitative studies, and texts whose full texts could not be accessed after three email contacts of principal investigators of the particular studies.

## Information source and search strategies

We used standardized and well-described methods in this systemic review [16]. Briefly, a search strategy was developed using fundamental concepts in the research question: Medical Subject Headings (MESH), keywords, and synonyms. The search strategy for PubMed: the keywords which we used in our search included terms describing stroke, age, and modifiable risk factors shown in the search strategy as follows: (1) (Stroke [Title] OR "Ischemic stroke"[-Title] OR "Ischaemic stroke"[Title] OR "Haemorrhagic stroke"[Title] OR "Hemorrhagic stroke" OR "Cerebral Vascular Accident" OR CVA); (2) (Adults OR "18 years or older") [Text Word] (3) (Ethiopia) [Text Word] (4) (Hypertension OR "High blood pressure" [Text Word] OR Diabetes [Text Word] OR "Diabetes mellitus" OR "Smoking" OR "Obesity" OR Alcohol OR "Heavy drinking" [Text Word] OR Physical exercise OR "Physical activity" [Text Word] OR (High blood cholesterol level OR "Hypercholesterolemia, Hyperlipidemia" OR "Hyperlipoproteinemia" OR "Arterial fibrillation") [Text Word] (5) #1 AND #2 AND #3 AND #4 (S1 Table).

A pretest of the search strategy by two authors was performed in PubMed. The actual electronic search was done from November 20 to 25, 2020. Two independent authors were implemented the electronic search in the following electronic databases: CINAHL (EBSCO), MEDLINE (via Ovid), PubMed, EMcare, AJOL, and Google scholar search engines. Finally, the search process was presented in a PRISMA flow chart.

## Study selection

Two of the reviewers (TWA and BWA) screened the titles and abstracts of each article to find potentially eligible studies. After removing duplicates, the search results were exported to End-Note software (version X7 Thomson Reuters, New York, NY) to create a bibliographical database of the retrieved references. The selection process was conducted in two stages: first screening of titles and abstracts against the predetermined inclusion/exclusion criteria, followed by a second screening of the full text of the research reports identified as probably relevant in the initial screening. Both stages were carried out independently by two authors (TWA and AG), and disagreements were resolved by discussion with another author (BWA).

## Data extraction process and quality assessment

The abstract and full-text review data abstraction was done by three independent authors (TWA, BZ, and AG) using a pre-piloted data extraction format prepared in the Microsoft™ Excel spreadsheet. Disagreement in data abstraction between the first two and third authors was resolved by a fourth independent author (BWA). From each observational study, we had extracted data regarding participant gender, study year, region, sample size, study design, and first author name. In addition to these data, the proportion of ischemic stroke, hemorrhagic stroke, improvement at discharge, and each modifiable risk factor (hypertension, diabetes

mellitus, alcohol consumption, smoking, heart disease, lack of physical activities, cholesterol, and obesity) was also extracted from each primary study.

Before analysis, prevalence transformation was carried out. The Newcastle-Ottawa Scale (NOS) was used to assess the quality of the included studies. The NOS had three categorical criteria with a maximum score of ten points. The assessment tool contains representatives of the sample, sample size, non-respondents, and ascertainment of exposure, independent blind assessment, and statistical test. Based on NOS, a score of 6 out of 10 was considered as good quality. To maintain the validity of this review, we only included primary studies with fair to good quality [17–19].

The primary outcome of this study was the pooled overall burden of stroke and its modifiable risk factors among stroke patients in Ethiopia. Stroke was defined as rapidly developing clinical signs of focal, or at times, global disturbance of cerebral function, lasting more than 24 hours or leading to death with no apparent cause other than a vascular origin [20, 21].

## Quality assessment

The risk of bias of included studies was assessed using the 10-item rating scale developed by Hoy et al. for prevalence studies [22]. The assessment tool has a representative sample size, data collection method, reliability, and validity of study tools, case definition, and prevalence periods of the studies. Researchers categorized each observational article study as having a low risk of bias ("yes" answers to domain questions) or a high risk of bias ("no" answers to domain questions). Each study was assigned a score of 1 (Yes) or 0 (No) for each domain, and these domain scores added to give an overall study quality score. Scores of 8–10 were considered as having a "low risk of bias," 6–7 a "moderate risk," and 0–5 a "high risk." For the least risk of bias classification, discrepancies between the reviewers resolved via consensus.

## Data analysis

**Heterogeneity test and publication bias.** Heterogeneity between the findings of the primary studies was assessed by using Cochran's Q test and quantified with the I-square statistics. A P-value of less than 0.1 was considered to suggest statistically significant heterogeneity. A heterogeneity was considered a small number of studies and their heterogeneity in design [23]. Heterogeneity classifications were: I-square values below 25% low, 25–75% moderate, and above 75% high [24]. Thus, the random-effect model was used to pool the burden of stroke and its modifiable risk factors since the studies were found heterogeneous [25].

We used the random-effect model to investigate the source of heterogeneity. The meta-analysis was weighted to account for the residual between-study heterogeneity (i.e., heterogeneity not explained by the covariate in the regression [26]. Publication bias was assessed by visual inspection of funnel plots based on the shape of the graph (subjective assessment). The symmetrical graph was interpreted to suggest an absence of publication bias, whereas an asymmetrical one indicated the presence of publication bias.

We employed Begg's and Egger's weighted regression to identify the source of publication bias (objective assessment). P-values less than 0.05 were considered as the presence of significant publication bias [27, 28]. We also applied a leave-out sensitivity analysis to estimate whether the pooled effect size was affected by a single studies. A leave-one-out sensitivity analysis was performed to confirm whether there were study potentially biased the direction of the pooled estimate. Subgroup analyses by region and type of study setup (hospitals) was carried out because of significant heterogeneity between studies (i.e., $I^2$ = 96.5%, p<0.05).

**Statistical analysis.** Data was analyzed in Stata Version 14. Data was presented in the evidence table and summarized using descriptive statistics. The effect measure for outcome

variables was computed using the "Metaprop" command for meta-analysis of the proportion in Stata. In this review, the overall burden of stroke, rate of improvement, and common modifiable risk factors were calculated together with their corresponding 95% CI. A forest plot was generated to display the pooled burden of strokes and its common modifiable risk stroke at 95% CI, the author's name, study year, and study weights.

# Result

## Study selection process

From electronic databases, we retrieved 986 observational studies. After screening their titles and abstracts, 644 duplications were removed using Endnote X7. Of the remaining 342 articles, 315 articles were excluded because their titles and abstracts were not in line with our inclusion criteria (full article not found, different population, different setting, and different outcome). Finally, 27 articles were included for this systemic review and meta-analysis (Fig 1).

## Study characteristics

Overall, we selected a total of 27 observational studies in this systematic review and meta-analysis. We included a total of 5,845 participants. Among them, 2,647 participants were male, and 3,228 participants improved at the time of discharge. The number of participants in each study ranged from 73 to 503. The most retrieved studies (n = 8) were from Oromia [29–36] followed by Addis Ababa (n = 7) [37–43], Amhara (n = 7) [44–50], Tigray region (n = 4) [51–54], and Southern Nations Nationalities and People's (SNNP) (n = 1) [55].

The smallest sample size was 73 obtained from a study conducted at Shashemene Referral Hospital, Ethiopia [29]. The largest sample size was 503 reported from a study done at Ayder Comprehensive Specialized Hospital, Northern Ethiopia [52]. Most studies dealt with hypertension as a modifiable risk factor of stroke (n = 24) [29–34, 36, 37, 38, 39–41, 43–46, 48, 50–55] followed by Atrial Fibrillation (AF) (n = 17) [30–32, 37, 38, 40, 41, 44–50–55], Diabetes mellitus (DM) (n = 15) [30, 31, 33, 37, 39–41, 44–46, 48, 49, 55–57], heart disease other than AF (n = 14) [29, 30, 32, 37, 38, 41, 43, 45–47, 49, 50, 54, 55], and high cholesterol levels (n = 7) [32, 33, 35, 37, 44, 48, 49, 51, 55] (Table 1).

## Quality appraisal

The quality score of the included study ranged from 5 to 8 to a mean score of 7.04 (SD = 0.94). Out of 27 studies, 21 (77.78%) studies received a low risk of bias. 5 studies [29–31, 36, 39, 44, 49, 51, 55] had a high risk of case definition, five studies [29, 30, 35, 45, 55] had random selection bias, and 14 studies [29, 33–35, 39, 41, 46, 47, 51, 53, 56] had a high risk of representation bias (S2 Table).

## The magnitude of strokes in Ethiopia

From the total rank of twenty-seven primary studies, twenty-five studies provided information on the proportion of hemorrhagic stroke. Twenty-six studies also provide information on stroke proportion in females and males. Twenty primary studies reported the rate of improvement at discharge after stroke. As presented in the forest plot (Figs 2 and 3), the pooled estimate proportion of hemorrhagic and ischemic stroke were 46.42% (95%CI: 41.82–51.53; $I^2$ = 91.6%) and 51.40% (95%CI: 46.97–55.82; $I^2$ = 85.5%) respectively. The pooled estimate of stroke among females was 45.07% (95%CI: 41.80–48.35; $I^2$ = 80.3%) and males was 54.70% (95%CI: 51.32–58.08; $I^2$ = 79.5%) (S1 File).

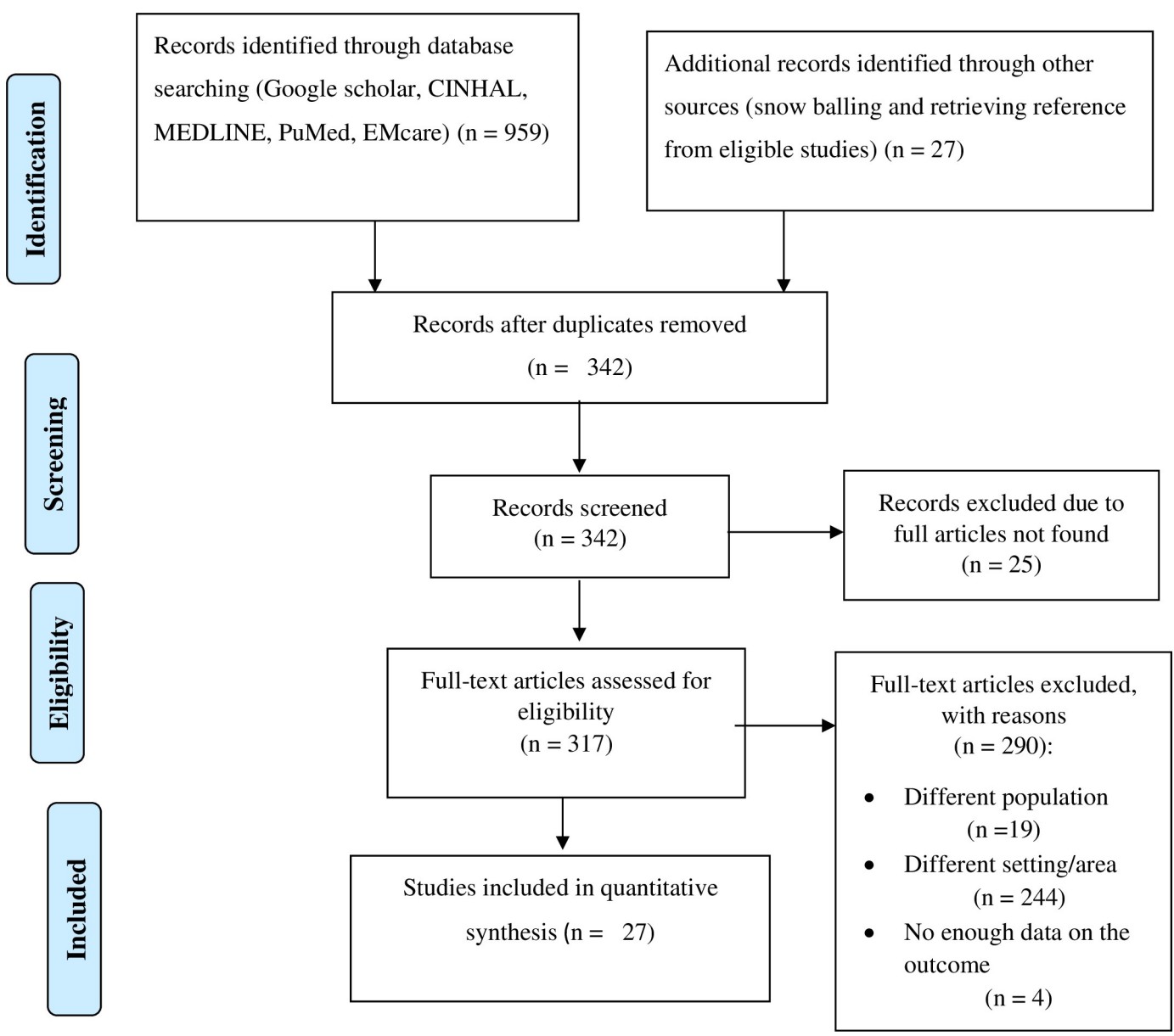

**Fig 1. Flow chart to a selection of studies for a systematic review and meta-analysis of the proportion of adherence to healthy lifestyle modification of people with hypertension in Ethiopia 2020.**

## The magnitude of modifiable risk factors of stroke in Ethiopia

We investigated the magnitude of modifiable risk factors of stroke among the included studies. The proportion of DM among stroke patients ranged from 5.2% [54] to 21.6% [39]. To estimate the magnitude of DM among stroke patient, we used a total of 3356 stroke patients. Accordingly, our pooled analysis showed that 14.722% (95%CI: 9.51, 19.94; $I^2$ = 95.8) of stroke patients had DM. In this review, stroke patients who had hypertension ranged from 24.1% [33] to 75.2% [32].

We studied a total of 5064 stroke patients to determine the pooled magnitude of hypertension in stroke patients. Consequently, we found that the overall pooled estimation of

**Table 1. Study characteristics of included articles for the final systematic review and meta-analysis on the burden of modifiable risk factors and rate of improvement at discharge after stroke in Ethiopia 2020.**

| Authors name | Study year | Region | Sample size | Study design | Burden of Stroke reported outcome percentage (95% CI) | | | | | |
|---|---|---|---|---|---|---|---|---|---|---|
| | | | | | Ischemic stroke | Hemorrhagic stroke | Female | Male | Improvement at discharge | NOS score |
| Asgedome SW.et al | 2019 | Tigray | 216 | R | 55.6 | 44.4 | 58.3 | 41.7 | 77.8 | 8 |
| Asres AK. et al | 2018 | AA | 170 | CC | 51.2 | 37.6 | 42.9 | 57.1 | 72.4 | 7 |
| Baye M. et al | 2018 | Amhara | 448 | R | 31.5 | 68.5 | 58.0 | 42.0 | 59.8 | 8 |
| Bedassa T. et al | 2018 | Oromia | 242 | R | 64.3 | 35.7 | -* | -* | -* | 5 |
| Beyene DT. et al | 2017 | Oromia | 367 | R | 35.7 | 64.31 | 36.2 | 63.8 | 26.4 | 8 |
| Dandena A. et al | 2019 | Oromia | 283 | P | 43.1 | 44.5 | 35.0 | 65.0 | -* | 6 |
| Deresse B. et al | 2014 | SNNP | 163 | P | 50.3 | 49.7 | 33.7 | 66.3 | 85.3 | 8 |
| Erkabu SG. et al | 2016 | Amhara | 303 | R | 59.4 | 40.6 | 37 | 63.0 | 89.0 | 7 |
| Fekadu G.et al | 2017 | Oromia | 116 | P | 51.7 | 48.3 | 37.1 | 62.9 | -* | 6 |
| Fekadu G.et al | 2017 | Oromia | 116 | CC | 48.3 | 41.6 | 37.1 | 62.9 | 78.4 | 8 |
| Fekadu G.et al | 2017 | Oromia | 364 | CC | 42.3 | 57.7 | 42.9 | 57.7 | 94.0 | 7 |
| Gebremariam SA. et al | 2014 | Tigray | 142 | CC | 55.6 | 38.0 | 45.8 | 54.2 | 47.9 | 8 |
| Gebreyohannes EA. et al | 2017 | Amhara | 208 | R | 57.7 | Not | 57.7 | 42.3 | 87.5 | 7 |
| Gedefa B. et al | 2016 | AA | 163 | R | 35.6 | 64.4 | 43.6 | 56.4 | 69.9 | 8 |
| Gelan Y. et al | 2016 | AA | 227 | CC | 49.8 | 48.9 | 30.0 | 70.0 | 70.0 | 7 |
| Greffie. ES et al | 2013 | Amhara | 98 | R | 69.4 | 30.6 | 53.1 | 46.9 | 87.0 | 7 |
| Gufue ZH. et al | 2019 | Tigray | 503 | R | 56.6 | 43.4 | 50.1 | 49.9 | 85.1 | 7 |
| Kassaw A.et al | 2018 | AA | 170 | R | 51.2 | 48.8 | 42.9 | 57.1 | 80.0 | 8 |
| Kefale B. et al | 2019 | Oromia | 111 | R | 80.1 | 18.0 | 50.5 | 49.5 | 83.8 | 7 |
| Mekonen HH.et al | 2018 | Tigray | 89 | R | 32.6 | 36.6 | 63.2 | 51.7 | -* | 5 |
| Mulat B. et al | 2015 | Amhara | 427 | R | 56.7 | 43.3 | 63.2 | 36.8 | -* | 6 |
| Mulugeta H. et al | 2019 | Amhara | 162 | R | 50.0 | 30.0 | 53.7 | 46.3 | 27.2 | 7 |
| Sultan M. et al | 2014 | AA | 301 | p | 53.8 | 17.9 | 42.5 | 57.5 | 80.7 | 8 |
| Tamirat KS. et al | 2017 | Amhara | 151 | R | 60.3 | 39.7 | 50.3 | 49.7 | 90.7 | 7 |
| Temesgen TG.et al | 2017 | Oromia | 73 | R | 65.8 | 34.2 | 42.5 | 57.5 | 54.8 | 6 |
| Zenebe G. et al | 2001 | AA | 128 | CC | 43 | 57.0 | 39.8 | 61.7 | -* | 6 |
| Zewdie A. et al | 2016 | AA | 104 | CC | 44.2 | 55.8 | 44.0 | 56.0 | -* | 5 |

R: Retrospective, P: Prospective, CC: Cross-Sectional, AA: Addis Ababa, NOS: Newcastle-Ottawa Scale

-*: The variable was not reported in the primary study.

hypertension among stroke patients was 49% (95CI%:43.59, 54.41; $I^2$ = 91.6%). Furthermore, the proportion of alcohol consumption (more than two drinks in a day for men and more than one drink in a day for women) among stroke patients included in this study ranged from 10.4% (55) to 41.4% (48). Our meta-analysis revealed that 24.96% (95%CI: 15.01, 34.90; $I^2$ = 92.7) of stroke patients had a history of harmful alcohol intake (Table 2).

## Recovery from stroke in Ethiopia

The proportion of improvement during discharge after stroke among the included primary studies was ranged from 26.4% [30] to 94% [36]. We included 2321 stroke patients to estimate the pooled proportion of improvement at the time of discharge. The pooled improvement status of stroke during discharge in Ethiopia was 72.28% (95%CI: 62.48, 82.08; $I^2$ = 96.5%) (Fig 4).

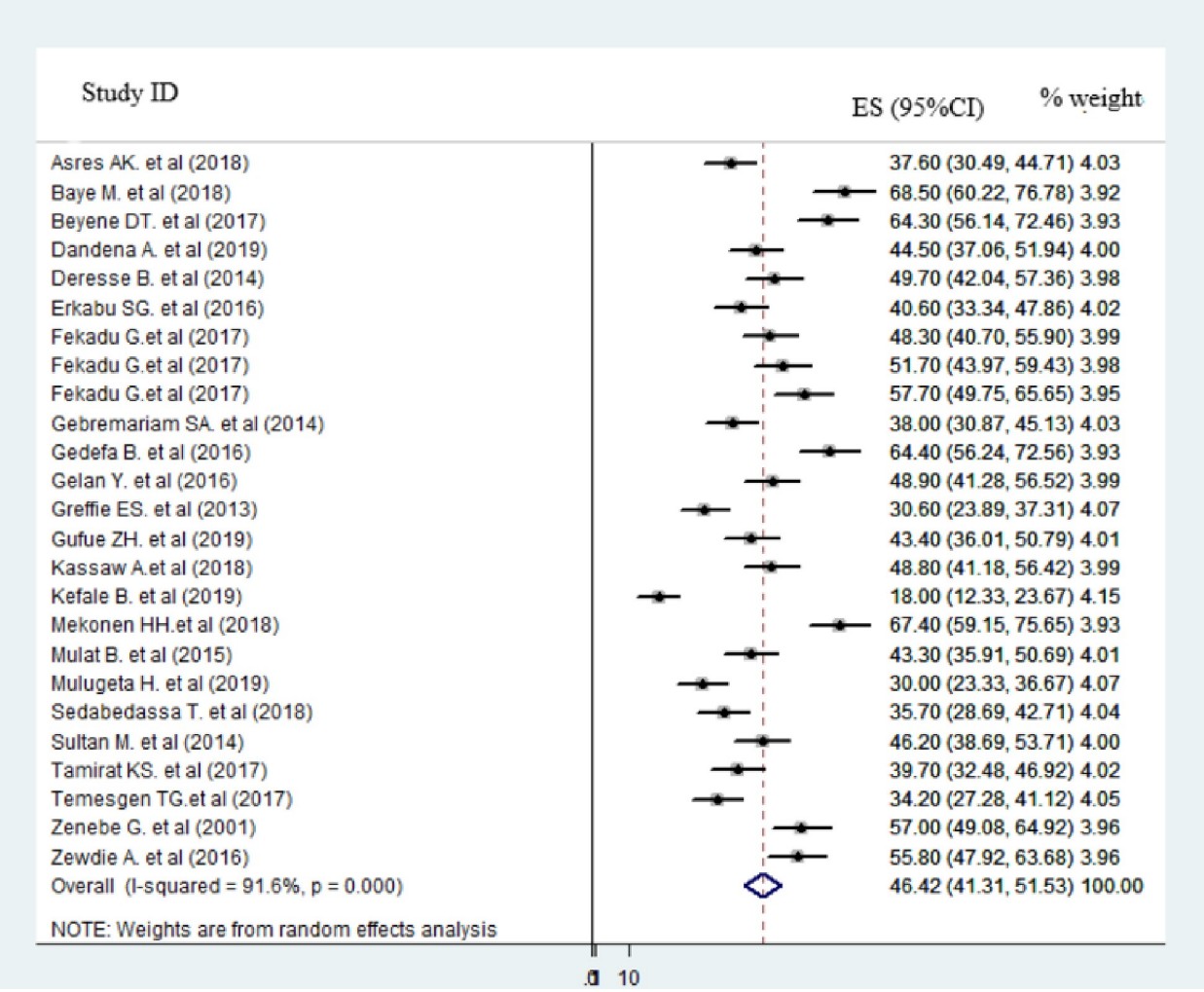

**Fig 2. Forest plot of in the proportion of hemorrhagic stroke in Ethiopia, 2020.**

## Publication bias

Both funnel plots of precision asymmetry and Egger's intercept test showed no publication bias in the primary studies. Visual examination of the funnel plot showed symmetric distribution. Additionally, Egger's intercept test was -0.147 (95% CI: -0.26, 1.18) p > 0.05 (0.102), and as judged by Egger's test, there was no evidence of publication bias present at a 5% significance level (Fig 5).

## Subgroup analysis

Due to the heterogeneity of included studies, we performed a subgroup analysis using the following study characteristics: region, sample size, and study year. We applied the random-effect model for reporting the pooled proportion of clinical outcomes during discharge in the subgroup analysis. Accordingly, the highest recovery rate (74.51) was observed from the Addis Ababa region (69.84–79.17; $I^2$ = 34.5%). The least pooled proportion of recovery (67.38) was in the Oromia region (95%CI: 41.60–93.17; $I^2$ = 98.1%). The subgroup analysis by study year showed that the pooled proportion of recovery rate after stroke during discharge was 75.59% (95%CI 64.28–86.9; $I^2$ = 92.1%) for studies conducted before 2017 (Table 3).

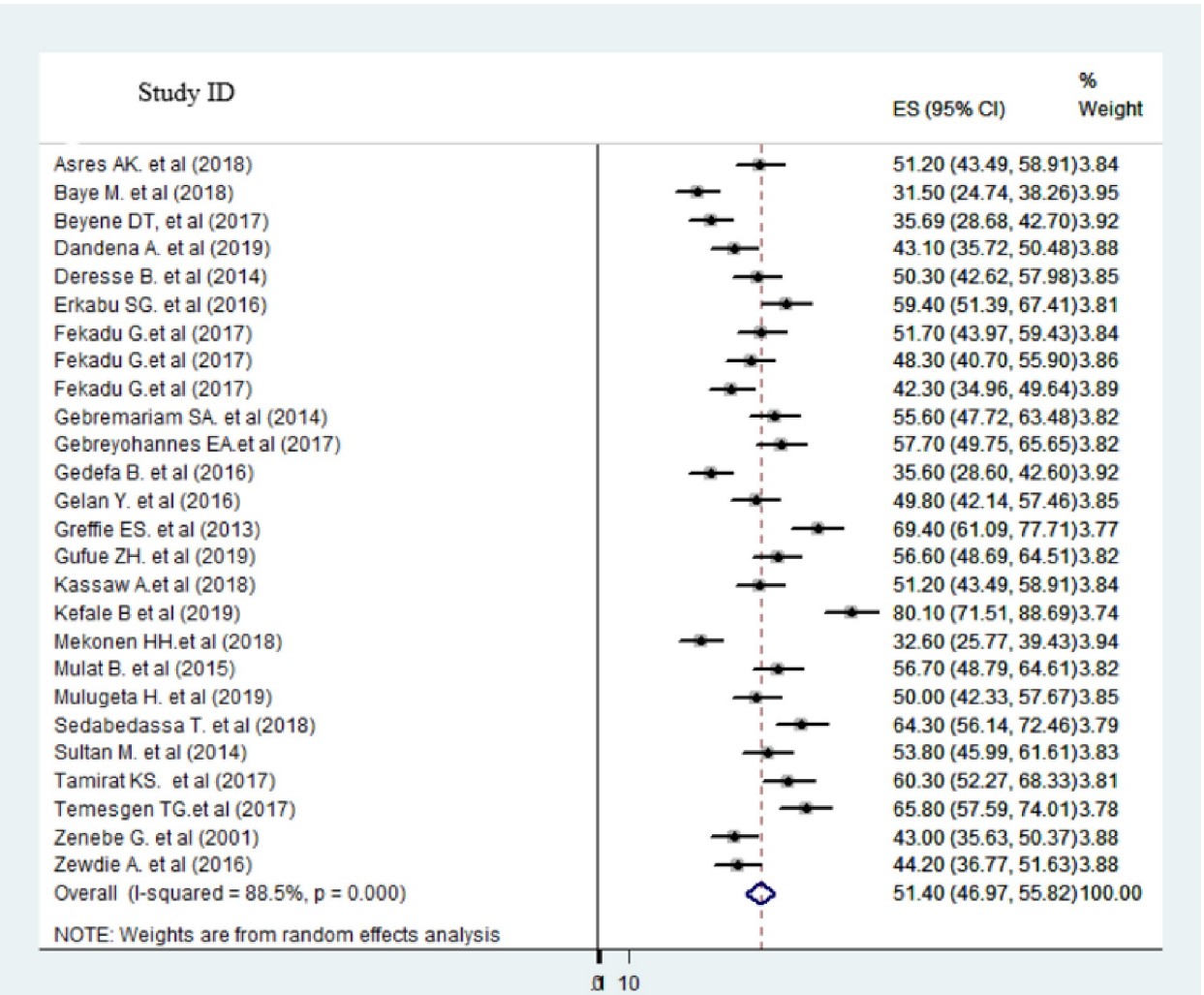

**Fig 3. Forest plot of in the proportion of ischemic stroke in Ethiopia, 2020.**

**Table 2. The pooled effect of common modifiable risk factors among the primary studies of stroke in Ethiopia.**

| Modifiable risk factors | Estimated pooled proportion (95%CI) | I-squared (%) |
|---|---|---|
| Hypertension | 49 (43.59, 54.41) | 91.6 |
| Diabetes mellitus | 14.72 (9.51, 19.94) | 95.8 |
| Atrial fibrillation | 19.21 (13.96, 24.46) | 94.4 |
| Other heart disease | 20.11 (14.27, 25.95) | 94.2 |
| Dyslipidemia | 20.99 (11.10, 30.88) | 96.4 |
| Smoking | 10.38 (6.27, 14.94) | 86.0 |
| Obesity | 11.64 (2.48, 20.79) | 95.3 |
| Alcohol | 24.96 (15.01, 34.90) | 92.7 |

Other heart Disease: Congestive heart failure, Structural heart disease, Myocardia friction.

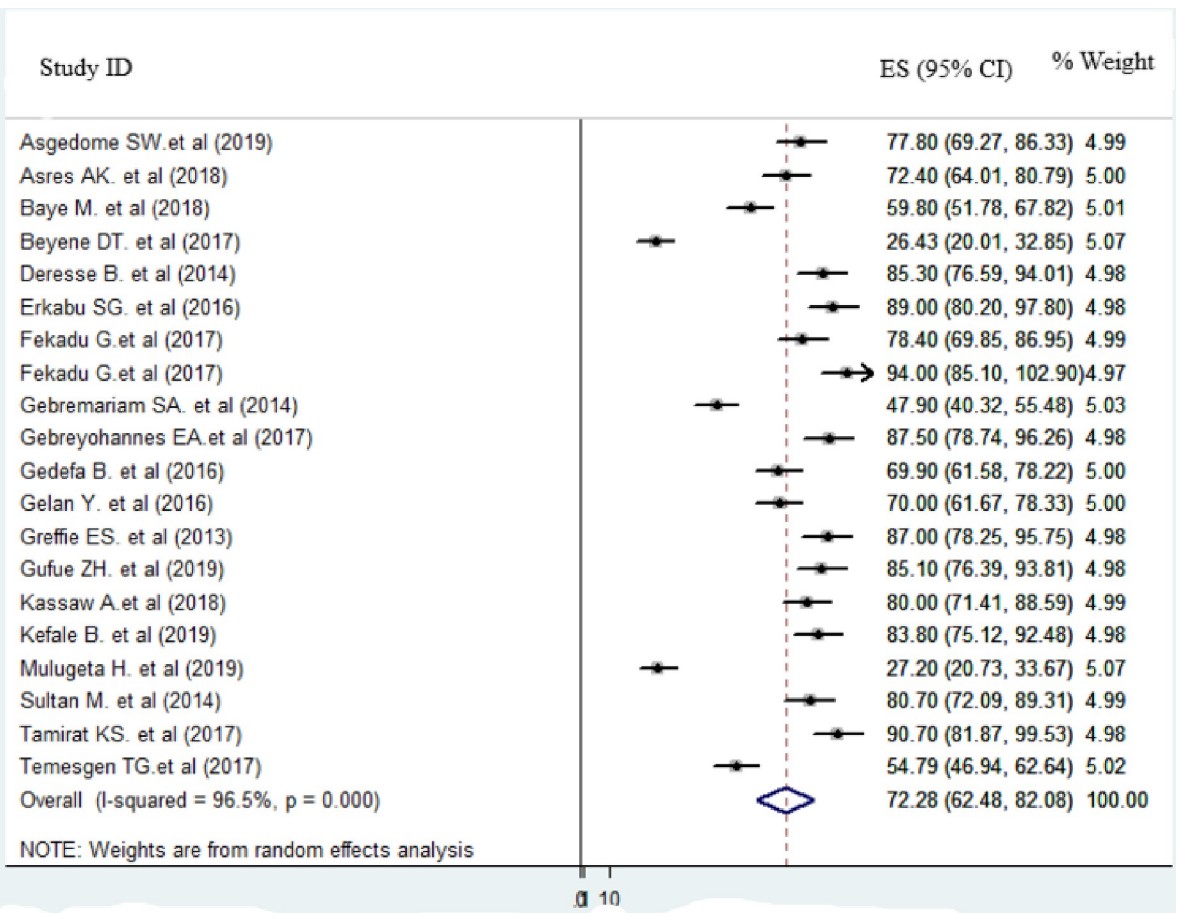

**Fig 4. Forest plot of the proportion of recover during discharge after stroke in Ethiopia, 2020.**

## Meta-regression and sensitivity analysis

The subgroup analysis showed that heterogeneity across the studies was widespread. To identify the source of heterogeneity, we conducted a meta-regression and sensitivity analysis. During the meta-regression analysis, we applied the following study covariance: study years and region. However, the results showed that none of these variables were a statistically significant source of heterogeneity. We also performed a sensitivity analysis to find the influence of each study on the overall effect size. No single study affected the overall pooled proportion of clinical outcomes of stroke among stroke patients in Ethiopia (Table 4, Fig 6).

## Discussion

This study aimed to determine the overall proportion of stroke burden and modifiable risk factors in Ethiopia. Of all stroke cases in our review, more than half (51.40%) of stroke patients in Ethiopia had ischemic subtype of stroke. While this finding was similar to study in Kenya (56.1%) [56]. It was much lower when compared to studies conducted in China (81.9–91.7%) [57, 58], Burkina Faso (61.63%) [59], Iran (76.5–81.9%) [60, 61], and a 22 countries case-control study (78%) [13]. The difference in culture and economic status, lifestyle difference, poor management of modifiable risk factors, and difference in the preventive strategies in the general public could be the reasons for the difference.

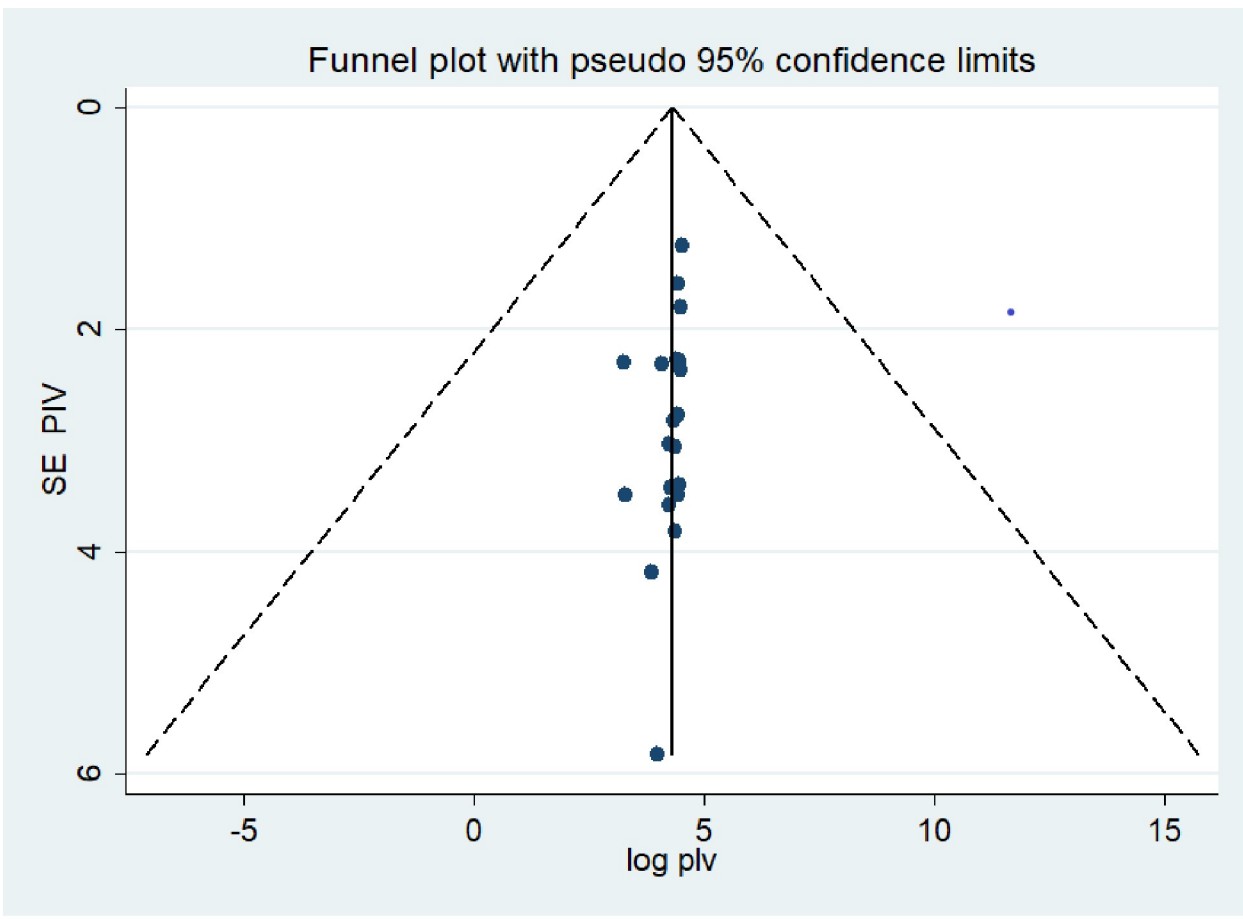

**Fig 5. Meta funnels presentations of the proportion of recover after stroke in Ethiopia, 2020, whereby SE PIV (standard error of proportion) plotted on the Y-axis and log PIV (logarithm of proportion).**

In this study, a higher prevalence of stroke was observed in males (54.70%) as compared to females (45.07%). A systematic review of epidemiological studies on Western European surveys has shown similar results with stroke being more common in males than females [62]. This gender difference is a hormonal makeup. The male sex is a known risk factor for stroke in

**Table 3. Subgroup analysis of recovery after stroke by region, sample size, and study year in Ethiopia 2020.**

| Variables | Characteristics | Estimated stroke recover during discharge (95% CI; $I^2$ = %) |
|---|---|---|
| Region | Oromia | 67.38 (41.60–93.17; $I^2$ = 98.1) |
| | Addis Ababa | 74.51 (69.84–79.17; $I^2$ = 34.5) |
| | Amhara | 73.44 (50.29–96. 59; $I^2$ = 97.9) |
| | Tigray | 70.19 (47.10–93.28; $I^2$ = 95.7) |
| | SNNPR | Single study |
| Sample size | <223 (median) | 72.41 (60.00–83.82; $I^2$ = 96.0) |
| | > = 223 (median) | 72.05 (527.44–91.66; $I^2$ = 97.5) |
| Study year | Before 2017 | 75.59 (64.28–86.9; $I^2$ = 92.1) |
| | After 2017 | 70.50 (56.80–84.20; $I^2$ = 92.1) |

SNNPR: South Nations, Nationalities and People Region.

**Table 4.  Meta-regression output to explore the heterogeneity of the pooled proportion of clinical outcome of stroke in Ethiopia, 2020.**

| Variables | Coefficients | P-value | 95% CI |
|---|---|---|---|
| Study Year | -4.83 | 0.538 | -21.03, 11.35 |
| Region | | | |
| Addis Ababa | -10.68 | 0.660 | -61.37, 40.01 |
| Amhara | -11.67 | 0.626 | -61.63, 38.29 |
| Oromia | -17.72 | 0.468 | -68.44, 33.01 |
| Tigray | -14.83 | 0.563 | -68.26, 38.61 |

humans, and female progesterone has a neuroprotective role in stroke [63]. There are clear differences in body size and vascular anatomy that are associated with an increased risk of stroke in males [64]. But females suffer from stroke at older ages making them more prone to die from stroke than males [65].

Our meta-analysis showed that almost half (49%) of all stroke patients had hypertension. Previous evidence has also shown that 75.8% of stroke patients had hypertension [59], hypertensive individuals are two to four times more likely to have a stroke [13, 57, 66]. Hypertension has remained the leading modifiable risk factor of stroke morbidity and mortality since 1990 [67]. People who can maintain normal blood pressure can decrease the risk of stroke by 30 to 40% [68].

Though hypertension is the main reported modifiable risk factor of stroke among the included primary studies, the pooled proportion of hypertension among stroke patients found in the current study is lower than the previous studies conducted in Burkina Faso [59], Iran [61], China [58], Bosnia-Herzegovina [69], Nigeria [70], and Bangladesh [71]. The possible explanation for this variation might be due to the lack of diagnostic modalities and proficiency, level of income, hypertension awareness, treatment, and control [72].

Above limit, alcohol consumption is a well-established risk factor of stroke. In our review, alcohol consumption is the second most common modifiable risk factor of stroke. Almost one-fourth (24.96%) of stroke patients had a history of alcohol consumption. Because harmful amounts of alcohol intake can trigger AF–a type of irregular heartbeat. Atrial fibrillation increases the risk of stroke by five times because it can cause blood clots to form in the heart. If these clots move up into the brain, it can lead to stroke [73].

In a review of 84 studies of alcohol consumption and cardiovascular disease, alcohol consumption >60 g/day increased the risk of incident stroke by 62% as compared to abstinence from alcohol [74]. The pooled proportion of alcohol consumption among stroke patients in Ethiopia was higher than a study conducted in Nigeria [72]. The possible explanation for this variation might be the lack of diagnostic modalities and proficiency; measured dyslipidemia in the medical record before the occurrence of stroke. Another reason for this variation is the lack of an effective community action to control alcohol consumption in Ethiopia [75, 76].

In our study, dyslipidemia is the third most common modifiable risk factor of stroke. More than two-tenths (20.99%) of stroke patients had dyslipidemia. Dyslipidemia promotes cervical or coronary atherosclerosis, which predisposes to athero-thrombotic and cardio-embolic stroke [77]. Our review is comparable with a previous study conducted in Nigeria [70]. However, this estimated proportion of dyslipidemia among stroke patients is lower than a study conducted in Bosnia-Herzegovina [69], and China [58].

The reasons for the above results could be attributed to the following: first, the dramatic increases in the prevalence of many known risk factors for chronic diseases such as unhealthy lifestyles (decreased physical activity, smoking, alcohol consumption, and westernized diet)

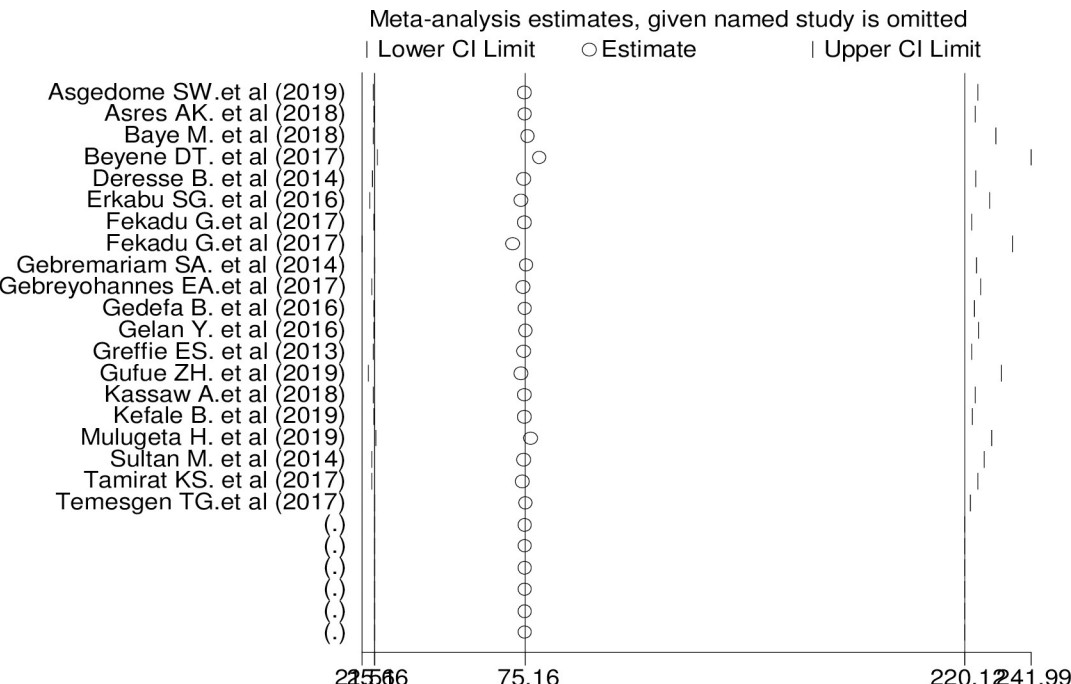

**Fig 6. One-leave-out sensitivity analysis for studies conducted on the pooled estimated proportion of stroke clinical outcome in Ethiopia, 2020.**

[78, 79]. Second, the impact of rapid urbanization (increased risk of obesity) [80]. In Nigeria, dyslipidemia in stroke patients is closely linked to western diet and physically inactive lifestyle behaviors [81].

Diabetes is a well-established risk factor for stroke, and our analysis showed that diabetes mellitus is the fourth most common comorbidity of stroke. More than one-tenth (14.72%) of stroke patients had diabetes mellitus. Diabetes causes various micro-vascular and macro-vascular changes ending in major clinical complications [82]. The findings are comparable to the previous studies conducted in Nigeria [70]; higher than the studies done in Burkina Faso [59], sub-Saharan African [83], and a systematic review and meta-analysis done in Ethiopia respectively [84]. However, this proportion of diabetes mellitus among stroke patients is lower than studies done in Bosnia-Herzegovina [69] and Iran [61].

The result found in this study showed that the overall pooled estimated proportion of recovery after stroke in Ethiopia was 72.28% (95%CI: 62.48, 82.08). This finding is in line with the ideal proportional recovery rule of stroke [85–87] but lower than the goal set for 2015 (85%) in the Helsingborg Declaration 2006 [88]. This result is still lower than the previous studies conducted in Bosnia-Herzegovina [69], Iran [61], sub-Saharan Africa [83], and Kenya [89]. The possible reason might be a sup-optimal management protocol for stroke patients and lack of skilled personnel, appropriate treatment, and diagnostic agents in Ethiopia [90]. However, this result is relatively higher than previous studies conducted in Ghana [91].

The subgroup analyses by year of studies showed that the overall pooled proportion of recovery rate after stroke was higher among studies conducted before 2017. The lowest pooled proportion of recovery rate of stroke in this study population after 2017 may reflect the increased exposure to risk factors for stroke due to ongoing epidemiological and demographic transitions.

### Limitations

There is considerable heterogeneity across the included studies. The observed heterogeneity may be attributed to differences in the study design, the quality of the studies, and sensitivity. Since our study focused on in-patient, it cannot externally validate to the general population.

### Implication

This study has many implications for clinical practice and future research. First, develop effective strategies to practice healthy life habit to prevent stroke burden. Second, there has been an increasing emphasis on the need for stroke services managed in the health care service, the community and rehabilitations service. Third, identifying the challenges to amend modifiable stroke risk factors is the first step in developing evidence-based interventions to promote short and long-term health outcomes and quality of life. Future research should focus on developing and testing a conceptual model that can use accessibility to screening, treatment, sociocultural aspects of stroke risk factor modification in a national context. Finally, to give a long-term reduction in burden of stroke and modifiable risk factor-related co-morbidity, researchers should assess ways to extend and sustain lifestyle modifiable risk factors and recovery rate after in this population.

## Conclusion

There is a high burden of stroke with a high rate of modifiable risk factors in Ethiopia. More than 90% of patients had one or more modifiable risk factors. Therefore, efforts should be focused on the primary prevention of stroke. Efforts should be taken to lower blood pressure, limit alcohol intake, early screen and treatment of atrial fibrillation and diabetes timely, quit smoking and improve physical activity.

## Supporting information

**S1 Checklist. PRISMA check list.**
(DOCX)

**S1 File. Figs 1 and 2.** Forest plot of in the proportion of stroke among female and male in Ethiopia, 2020.
(DOCX)

**S1 Table. Search strategy applied to PubMed database in the current review.**
(DOCX)

**S2 Table. Risk of bias assessment tool of eligible articles by using the Hoy 2012 tool.**
(DOCX)

**S3 Table. Scoring of the quality of articles by authors using the Newcastle-Ottawa quality assessment tool.**
(XLSX)

**S4 Table. Data extraction speared sheet.**
(XLSX)

## Author Contributions

**Conceptualization:** Teshager Weldegiorgis Abate.

**Data curation:** Teshager Weldegiorgis Abate, Balew Zeleke, Ashenafi Genanew, Bidiru Weldegiorgis Abate.

**Formal analysis:** Teshager Weldegiorgis Abate, Balew Zeleke, Ashenafi Genanew, Bidiru Weldegiorgis Abate.

**Methodology:** Teshager Weldegiorgis Abate, Balew Zeleke.

**Software:** Teshager Weldegiorgis Abate.

**Writing – original draft:** Teshager Weldegiorgis Abate, Bidiru Weldegiorgis Abate.

**Writing – review & editing:** Teshager Weldegiorgis Abate, Balew Zeleke, Ashenafi Genanew, Bidiru Weldegiorgis Abate.

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
