## [Decision Letter · Decision Letter 0]

21 Apr 2021

PONE-D-21-01404

Patient recovery from stroke and modifiable risk factors in Ethiopia: a systemic review and meta-analysis.

PLOS ONE

Dear Dr. Abate,

Thank you for submitting your manuscript to PLOS ONE. After careful consideration, we feel that it has merit but does not fully meet PLOS ONE’s publication criteria as it currently stands. Therefore, we invite you to submit a revised version of the manuscript that addresses the points raised during the review process.

We look forward to receiving your revised manuscript.

Kind regards,

Miguel A. Barboza, MD, MSc

Academic Editor

PLOS ONE

Journal Requirements:

NO - Include this sentence at the end of your statement: The funders had no role in study design, data collection and analysis, decision to publish, or preparation of the manuscript.

2a)           Please clarify the sources of funding (financial or material support) for your study. List the grants or organizations that supported your study, including funding received from your institution.

2b)           State what role the funders took in the study. If the funders had no role in your study, please state: “The funders had no role in study design, data collection and analysis, decision to publish, or preparation of the manuscript.”

2C)           If any authors received a salary from any of your funders, please state which authors and which funders.

2d)           If you did not receive any funding for this study, please state: “The authors received no specific funding for this work.”

4. Thank you for submitting the above manuscript to PLOS ONE. During our internal evaluation of the manuscript, we found significant text overlap between your submission and the following previously published works, some of which you are an author.

https://bmcneurol.biomedcentral.com/articles/10.1186/s12883-020-01870-6

https://journals.plos.org/plosone/article?id=10.1371%2Fjournal.pone.0229698

https://bmcpublichealth.biomedcentral.com/articles/10.1186/s12889-019-7505-7

Please revise the manuscript to rephrase the duplicated text, cite your sources, and provide details as to how the current manuscript advances on previous work. Please note that further consideration is dependent on the submission of a manuscript that addresses these concerns about the overlap in text with published work.

Reviewers' comments:

Reviewer's Responses to Questions

**Comments to the Author**

1. Is the manuscript technically sound, and do the data support the conclusions?

Reviewer #1: Yes

Reviewer #2: Yes

Reviewer #3: Yes

2. Has the statistical analysis been performed appropriately and rigorously? 

Reviewer #1: Yes

Reviewer #2: No

Reviewer #3: I Don't Know

3. Have the authors made all data underlying the findings in their manuscript fully available?

Reviewer #1: Yes

Reviewer #2: Yes

Reviewer #3: Yes

4. Is the manuscript presented in an intelligible fashion and written in standard English?

Reviewer #1: No

Reviewer #2: No

Reviewer #3: No

5. Review Comments to the Author

Reviewer #1: This is a good effort by the authors highlighting the burden of modifiable risk factors in stroke. The methodology and analysis approach is commendable.

However there are few points the need consideration:

1. There are multiple language errors, grammatical mistakes, typing errors which alter the meaning of what the authors may be trying to convey.

2. There is use of different fonts in different sections of the manuscript, which the authors need to address.

3. The discussion section needs to highlight the reasons for the difference in results of each significant risk factor when compared to other studies quoted.

4. The conclusion should emphasize how this study adds to literature; authors should point out feasible strategies suggested by them that can help address the modifiable risk factors in Ethiopia.

Reviewer #2: The authors tried to review stroke outcomes and modified risk factors in Ethiopia. I would like to thank the authors for trying to address the topic in caption.

I have the following comments for the authors, hoping these will make the article more plausible to readers:

1. Revise the research type as "review" not "research article"

2. Explain the operational definition you have used regarding "alcohol consumption", its broad term, is it any alcohol consumption?

3. You have not used "AJOL: database. However, most of articles from Ethiopia were published on local and regional journals; which are indexed by AJOL. So, I believe by not including AJOL in your searching process, you may miss many articles from Ethiopia.

4. You have also included un-published articles. But how did you access this un-published articles? because I know there are many published and un-published articles from our departments you haven't included.

5. The search key words are very few, to capture all modifiable stroke risk factor. Please explain this issue.

6. In the Analysis process, since the authors mentioned meta-analysis, they should present their results in "Forest plot", which they didn't included.

7. Please revise your discussion section by comparing your findings with other regions mainly focusing on points unique to this region.

8. Include your data extraction detail tables as a supplementary file for detail evaluation of your work.

Reviewer #3: Dear Editor,

Thanks for choosing me to review articles in your prestigious journal. I read the Research entitled: Patient recovery from stroke and modifiable risk factors in Ethiopia: a systemic review and meta-analysis. Here are some of my comments to the manuscript which needs from my point of view good English editing.

The Title: was unclear and ambiguous as one doesn’t understand whether the authors would investigate the impact of modifiable risk factors on recovery after stroke or would study both items in one research. It’s clear also that the patient in their results studied more items like magnitude of stroke in Ethiopia, stroke sub-types in Ethiopia, so this title should be changed to (for example): Burden of stroke in Ethiopia.

In the Abstract:

Background:

• Line 2,3 : The contribution of modifiable risk factors to the increasing global and regional burden of stroke is unclear. This sentence is not correct as the contribution of modifiable risk factors to the global burden of stroke is well studied in the international literature, so they should remove the word global from the sentence and specify the sentence to be regional (in Ethiopia).

• Line 3: crucial for informing stroke prevention strategies. This sentence is wrong, they should write for establishing stroke prevention strategies and I suggest that, the language of the paper should be revised by native speaker to be more clear and informative.

• Line 5: one of the keys to improving the quality of life. To improve not to improving, and as I said before the language of paper should be revised and rewritten by native speaker .

Conclusion: needs to be rewritten , it should be more concise and informative

Background:

• The introduction part was concise and clear, however as I said before need English editing.

• Page 3, Lines 17-20: should be removed as they are inconsistent with the research idea

• Page 4: Line 3 is a repetition to line 1, should be removed.

Methods and Analysis:

• Well written

The Results

Well written, however it needs like other manuscript parts English editing to be easier and more concise

• Magnitude of strokes in Ethiopia

The manuscript title was about the recovery from stroke and the modifiable risk factors, and in the results section there is big paragraph about the magnitude of stroke in Ethiopia without any hint in the abstract section or the title, this is should be considered by the authors, I recommend to change the title to for example: Burden of stroke in Ethiopia would be more informative

The discussion:

• The discussion part is poorly written. The authors should focus on comparing their findings with the similar publications from the same region, they should focus on African publications and when they compare with other international publications, there are many publications focusing on African American for example, but comparing their results with Chinese population is misleading as Asian population have distinct characters, distribution of risk factors and distinct pattern of atherosclerosis (intracranial mainly) which is different from African population.

6. PLOS authors have the option to publish the peer review history of their article (what does this mean?). If published, this will include your full peer review and any attached files.

Reviewer #1: **Yes: **Ivy Anne Sebastian

Reviewer #2: **Yes: **Biniyam A. Ayele, MD

Reviewer #3: No

---

## [Author Response · Author response to Decision Letter 0]

8 May 2021

Response to Reviewers

Response to editor and reviewers’

Response to the editor: 

We thank you and the reviewers for a thorough reading and constructive criticism of our manuscript and for the opportunity to revise and resubmit. We are pleased to submit the improved research article, including a proposed comment, “The burden of stroke and modifiable risk factors in Ethiopia: a systemic review and meta-analysis with a manuscript ID of PONE-D-21-01404” 

1. General Comments:

#1. COMMENT: Please ensure that your manuscript meets PLOS ONE's style requirements, including those for file naming.

RESPONSE: We have checked and attest that all formatting and style requirements have

been met PLOS ONE's style requirements. 

#2. COMMENT: The funders had no role in study design, data collection and analysis, decision to publish, or preparation of the manuscript. At this time, please address the following queries…

RESPONSE: The authors received no specific funding for this work.

#3. COMMENT: Please include captions for your Supporting Information files at the end of your manuscript, and update any in-text citations to match accordingly.

RESPONSE: We include all capitation 

#4. COMMENT: Thank you for submitting the above manuscript to PLOS ONE. During our internal evaluation of the manuscript, we found significant text overlap between your submission and the following previously published works, some of which you are an author.

RESPONSE: We revise the manuscript to rephrase the duplicated text.

2. Review Comments to the Author

REVIEWER #1 COMMENTS

1. COMMENT: There are multiple language errors, grammatical mistakes, typing errors which alter the meaning of what the authors may be trying to convey.

RESPONSE: the language expert copyedited the manuscript for language, spelling, grammar and sentence structure.

2. COMMENT: There is use of different fonts in different sections of the manuscript, which the authors need to address.

RESPONSE: Thank you this comment. We make uniform of fonts throughout the manuscript. 

3. COMMENT: The discussion section needs to highlight the reasons for the difference in results of each significant risk factor when compared to other studies quoted.

RESPONSE: We try to highlight the reason from the different results of each significant risk factor when compared to other studies quoted.

4. COMMENT: The conclusion should emphasize how this study adds to literature; authors should point out feasible strategies suggested by them that can help address the modifiable risk factors in Ethiopia.

RESPONSE: We accept the comment and we emphasize feasible strategies in the nation- Ethiopia. 

REVIEWER #2 COMMENTS

1. COMMENT: Revise the research type as "review" not "research article"

RESPONSE: we accept the comment and replace research article" by “review”

2. COMMENT: Explain the operational definition you have used regarding "alcohol consumption", its broad term, is it any alcohol consumption?

RESPONSE: when to say alcohol consumption, more than two drinks in a day for men and more than 1 drink in a day for women

3. COMMENT: You have not used "AJOL: database. However, most of articles from Ethiopia were published on local and regional journals; which are indexed by AJOL. So, I believe by not including AJOL in your searching process, you may miss many articles from Ethiopia.

RESPONSE: we included AJOL for a data bases

4. COMMENT: You have also included un-published articles. But how did you access this un-published articles? Because I know there are many published and un-published articles from our departments you haven't included.

RESPONSE: we access the institutional repository (like Jimma University, Addis Ababa University). If you have an included article, please share a link and we are ready to included 

5. COMMENT: The search key words are very few, to capture all modifiable stroke risk factor. Please explain this issue.

RESPONSE: we select the most public health important modifiable factors 

6. COMMENT: In the Analysis process, since the authors mentioned meta-analysis, they should present their results in "Forest plot", which they didn't included.

RESPONSE: we try to present the result in ‘Forest plot’ in figure 2 to figure 4.

7. COMMENT: Please revise your discussion section by comparing your findings with other regions mainly focusing on points unique to this region.

RESPONSE: we try to discuss by comparing our findings with other regions mainly focusing on points unique to this region.

8. COMMENT: Include your data extraction detail tables as a supplementary file for detail evaluation of your work.

RESPONSE: we include data extraction detail tables as a supplementary file.

REVIEWER #3 COMMENTS

1. COMMENT: The Title: was unclear and ambiguous as one doesn’t understand. So this title should be changed to (for example): Burden of stroke in Ethiopia.

RESPONSE: we changed the tittle as recommended “Burden of stroke and modifiable risk factors in Ethiopia: a systemic review and meta-analysis.”

On abstract part

2. COMMENT: abstract on back ground, line 2, 3. This sentence is not correct as the contribution of modifiable risk factors to the global burden of stroke is well studied in the international literature, so they should remove the word global from the sentence and specify the sentence to be regional (in Ethiopia).

RESPONSE: the sentence is specified in regional context-Ethiopia.

3. COMMENT: Back ground line 3: crucial for informing stroke prevention strategies. This sentence is wrong, they should write for establishing stroke prevention strategies and I suggest that, the language of the paper should be revised by native speaker to be more clear and informative. 

RESPONSE: we accept the comment and write as the reviewer suggestion. 

4. COMMENT: back ground line 5: one of the keys to improving the quality of life. To improve not to improving, and as I said before the language of paper should be revised and rewritten by native speaker 

RESPONSE: we accept the comment and write as the reviewer suggestion.

5. COMMENT: Conclusion: needs to be rewritten, it should be more concise and informative.

RESPONSE: we try to revised the conclusion section.

---

## [Decision Letter · Decision Letter 1]

9 Jun 2021

PONE-D-21-01404R1

The burden of stroke and modifiable risk factors in Ethiopia: a systemic review and meta-analysis.

PLOS ONE

Dear Dr. Abate,

Thank you for submitting your manuscript to PLOS ONE. After careful consideration, we feel that it has merit but does not fully meet PLOS ONE’s publication criteria as it currently stands. Therefore, we invite you to submit a revised version of the manuscript that addresses the points raised during the review process.

We look forward to receiving your revised manuscript.

Kind regards,

Miguel A. Barboza, MD, MSc

Academic Editor

PLOS ONE

Journal Requirements:

Additional Editor Comments (if provided):

Reviewers' comments:

Reviewer's Responses to Questions

**Comments to the Author**

1. If the authors have adequately addressed your comments raised in a previous round of review and you feel that this manuscript is now acceptable for publication, you may indicate that here to bypass the “Comments to the Author” section, enter your conflict of interest statement in the “Confidential to Editor” section, and submit your "Accept" recommendation.

Reviewer #1: (No Response)

Reviewer #3: All comments have been addressed

2. Is the manuscript technically sound, and do the data support the conclusions?

Reviewer #1: Yes

Reviewer #3: Yes

3. Has the statistical analysis been performed appropriately and rigorously? 

Reviewer #1: Yes

Reviewer #3: Yes

4. Have the authors made all data underlying the findings in their manuscript fully available?

Reviewer #1: Yes

Reviewer #3: Yes

5. Is the manuscript presented in an intelligible fashion and written in standard English?

Reviewer #1: No

Reviewer #3: Yes

6. Review Comments to the Author

Reviewer #1: 1. After reviewing the corrected manuscript, I would like to bring forward my concerns to the authors. Although the content of the paper is good, however the language errors and grammatical mistakes in the document are still far too many to consider this as a meaningful submission. The typing errors, missing verbs and punctuations completely take away from what the authors are trying to convey. I strongly recommend that they take the help of an English language expert to thoroughly revise the whole manuscript before submission.

2. In my previous review I had submitted an attachment with many comments and edits (apart from comments), however those have not been addressed at all. I urge the authors to go through the attachments submitted along with the comments as well before submitting their revisions.

3. The references are not in the correct format and there is no uniformity. Authors names as well as journal names are missing. Authors need to re-write this whole section and present the bibliography correctly.

4. I would like to re-iterate that the authenticity and content of a paper are just as important as the presentation, for acceptance to any journal. I request the authors to thoroughly go through the document again and make relevant edits before submission.

Reviewer #3: (No Response)

7. PLOS authors have the option to publish the peer review history of their article (what does this mean?). If published, this will include your full peer review and any attached files.

Reviewer #1: No

Reviewer #3: **Yes: **Ahmed Nasreldein

---

## [Author Response · Author response to Decision Letter 1]

23 Jun 2021

Response to Reviewers

Response to editor and reviewer

we thank you and the reviewers for a thorough reading and constructive criticism of our manuscript and for the opportunity to revise and resubmit. We are pleased to submit the improved research article, including a proposed comment, “The burden of stroke and modifiable risk factors in Ethiopia: a systemic review and meta-analysis.” 

RESPONSE TO REVIEWER 1

REVIEWER #1 COMMENT: 

1. Although the content of the paper is good, however the language errors and grammatical mistakes in the document are still far too many to consider this as a meaningful submission. The typing errors, missing verbs and punctuations completely take away from what the authors are trying to convey. I strongly recommend that they take the help of an English language expert to thoroughly revise the whole manuscript before submission.

RESPONSE: we try to address the comment and one fried who is a language expert help me in editorial technique. 

2. In my previous review I had submitted an attachment with many comments and edits (apart from comments), however those have not been addressed at all. I urge the authors to go through the attachments submitted along with the comments as well before submitting their revisions.

RESPONSE: we addressed all the important comment in the revised manuscript.

3. The references are not in the correct format and there is no uniformity. Authors’ names as well as journal names are missing. Authors need to re-write this whole section and present the bibliography correctly.

RESPONSE: we intensive edit the whole reference by using endnote reference manager software.

4. . I would like to re-iterate that the authenticity and content of a paper are just as important as the presentation, for acceptance to any journal. I request the authors to thoroughly go through the document again and make relevant edits before submission.

RESPONSE: we try to thoroughly edit and incorporate all the comment of the reviewer

---

## [Decision Letter · Decision Letter 2]

2 Aug 2021

PONE-D-21-01404R2

The burden of stroke and modifiable risk factors in Ethiopia: a systemic review and meta-analysis.

PLOS ONE

Dear Dr. Abate,

Thank you for submitting your manuscript to PLOS ONE. After careful consideration, we feel that it has merit but does not fully meet PLOS ONE’s publication criteria as it currently stands. Therefore, we invite you to submit a revised version of the manuscript that addresses the points raised during the review process.

We look forward to receiving your revised manuscript.

Kind regards,

Miguel A. Barboza, MD, MSc

Academic Editor

PLOS ONE

Journal Requirements:

Additional Editor Comments (if provided):

Reviewers' comments:

Reviewer's Responses to Questions

**Comments to the Author**

1. If the authors have adequately addressed your comments raised in a previous round of review and you feel that this manuscript is now acceptable for publication, you may indicate that here to bypass the “Comments to the Author” section, enter your conflict of interest statement in the “Confidential to Editor” section, and submit your "Accept" recommendation.

Reviewer #1: (No Response)

2. Is the manuscript technically sound, and do the data support the conclusions?

Reviewer #1: Yes

3. Has the statistical analysis been performed appropriately and rigorously? 

Reviewer #1: Yes

4. Have the authors made all data underlying the findings in their manuscript fully available?

Reviewer #1: Yes

5. Is the manuscript presented in an intelligible fashion and written in standard English?

Reviewer #1: Yes

6. Review Comments to the Author

Reviewer #1: The edited document is well-written and conveys this important topic well.

1. The references are however not according to format. Journal abbreviations need to be formatted.

2. Few other minor errors are present, which I have addressed in the attachment.

7. PLOS authors have the option to publish the peer review history of their article (what does this mean?). If published, this will include your full peer review and any attached files.

Reviewer #1: **Yes: **Ivy Sebastian

---

## [Author Response · Author response to Decision Letter 2]

12 Aug 2021

Response to editor and reviewer

We thank you and the reviewers for a thorough reading and constructive criticism of our manuscript and for the opportunity to revise and resubmit. We are pleased to submit the improved research article, including a proposed comment, “The burden of stroke and modifiable risk factors in Ethiopia: a systemic review and meta-analysis.” 

Response to editorial comment

Comment: Journal Requirements: Please review your reference list to ensure that it is complete and correct. If you have cited papers that have been retracted, please include the rationale for doing so in the manuscript text, or remove these references and replace them with relevant current references. Any changes to the reference list should be mentioned in the rebuttal letter that accompanies your revised manuscript. If you need to cite a retracted article, indicate the article’s retracted status in the References list and also include a citation and full reference for the retraction notice.

RESPONSE: We revised the reference list a cording to journal requirements. Reference ’75: Alcohol and stroke, Factsheet13 (2014). ‘ with ‘Zhang C, Qin Y-Y, Chen Q, Jiang H, Chen X-Z, Xu C-L, et al. Alcohol intake and risk of stroke: a dose–response meta-analysis of prospective studies. International journal of cardiology. 2014;174’ the reason of replacement, the update one more appropriate one.(669-677)’ b 

RESPONSE TO REVIEWER 1

REVIEWER #1 COMMENT: 

1. The references are however not according to format. Journal abbreviations need to be formatted.

RESPONSE: we used reference manager software (EndNote) and mange according to this software manger.

2. Few other minor errors are present, which I have addressed in the attachment.

RESPONSE: we addressed all comment, which we have found in the attachment comments.

---

## [Decision Letter · Decision Letter 3]

23 Sep 2021

PONE-D-21-01404R3The burden of stroke and modifiable risk factors in Ethiopia: a systemic review and meta-analysis.PLOS ONE

Dear Dr. Abate,

Thank you for submitting your manuscript to PLOS ONE. After careful consideration, we feel that it has merit but does not fully meet PLOS ONE’s publication criteria as it currently stands. Therefore, we invite you to submit a revised version of the manuscript that addresses the points raised during the review process. Please see minor suggestions from one of the reviewers.

We look forward to receiving your revised manuscript.

Kind regards,

Miguel A. Barboza, MD, MSc

Academic Editor

PLOS ONE

Journal Requirements:

Reviewers' comments:

Reviewer's Responses to Questions

**Comments to the Author**

1. If the authors have adequately addressed your comments raised in a previous round of review and you feel that this manuscript is now acceptable for publication, you may indicate that here to bypass the “Comments to the Author” section, enter your conflict of interest statement in the “Confidential to Editor” section, and submit your "Accept" recommendation.

Reviewer #1: (No Response)

2. Is the manuscript technically sound, and do the data support the conclusions?

Reviewer #1: Yes

3. Has the statistical analysis been performed appropriately and rigorously? 

Reviewer #1: Yes

4. Have the authors made all data underlying the findings in their manuscript fully available?

Reviewer #1: Yes

5. Is the manuscript presented in an intelligible fashion and written in standard English?

Reviewer #1: Yes

6. Review Comments to the Author

Reviewer #1: Please edit your references. The journal names should be in NLM abbreviated format.

For eg:

"Gorelick PB. The global burden of stroke: persistent and disabling. The Lancet Neurology.

22 2019;18(5):417-8." the correct NLM abbreviation would be Lancet Neurol.

All the references need to be edited in this format. If the authors are unaware, suggest to take help regarding the same before submitting.

7. PLOS authors have the option to publish the peer review history of their article (what does this mean?). If published, this will include your full peer review and any attached files.

Reviewer #1: **Yes: **Ivy Sebastian

---

## [Author Response · Author response to Decision Letter 3]

13 Oct 2021

Response to editor and reviewer

We thank you and the reviewers for a thorough reading and constructive criticism of our manuscript and for the opportunity to revise and resubmit. We are pleased to submit the improved research article, including a proposed comment, “The burden of stroke and modifiable risk factors in Ethiopia: a systemic review and meta-analysis.” 

RESPONSE TO EDITORIAL COMMENT AND RESPONSE TO REVIEWER 1

EDITORIAL/REVIEWER #1 COMMENT: 

RESPONSE: we used reference manager software (EndNote) and mange according to this software manger. And also rewrite with ‘NLM abbreviation forma’

2. Please edit your references. The journal names should be in NLM abbreviated format.

RESPONSE: we try to edit all reference in NLM abbreviation format.

---

## [Editor Report · Decision Letter 4]

18 Oct 2021

The burden of stroke and modifiable risk factors in Ethiopia: a systemic review and meta-analysis.

PONE-D-21-01404R4

Dear Dr. Abate,

We’re pleased to inform you that your manuscript has been judged scientifically suitable for publication and will be formally accepted for publication once it meets all outstanding technical requirements.

Kind regards,

Miguel A. Barboza, MD, MSc

Academic Editor

PLOS ONE
---

## [Editor Report · Acceptance letter]

20 Oct 2021

PONE-D-21-01404R4 

The burden of stroke and modifiable risk factors in Ethiopia: a systemic review and meta-analysis. 

Dear Dr. Abate:

I'm pleased to inform you that your manuscript has been deemed suitable for publication in PLOS ONE. Congratulations! Your manuscript is now with our production department. 

Kind regards, 

on behalf of

Dr. Miguel A. Barboza 

Academic Editor

PLOS ONE